# Digital Embodiment as a Tool for Constructing the Self in Politics

Vincenzo Auriemma [1,*], Daniele Battista [2] and Serena Quarta [1]

1 Dipartimento di Studi Politici e Sociali, Università degli Studi di Salerno, 84084 Salerno, SA, Italy; squarta@unisa.it
2 Dipartimento di Scienze Aziendali—Management & Innovation Systems/DISA-MIS, Università degli Studi di Salerno, 84084 Salerno, SA, Italy; dbattista@unisa.it
* Correspondence: vauriemma@unisa.it

**Abstract:** This article offers an exploration of the theoretical and methodological implications related to the concept of digital embodiment in the field of contemporary communication. It seeks to analyze a crucial intersection between the virtual and material dimensions of human experience, enabling a deeper understanding of how bodies are shaped, visualized, and experienced in the digital age. Specifically, through this conceptual lens, we examine how human bodies are engaged in continuous interaction with digital technologies, giving rise to new forms of presence and identity. Therefore, we will seek to analyze how the personalization of the body within political communication has been profoundly affected by the virtualization of human experience. Next, we will introduce a new approach, useful for studying this fusion, that can emphasize the importance of analyzing this issue using ethnography, which is useful for fully understanding the complex dynamics surrounding the personalized digital body.

**Keywords:** digital embodiment; co-construction of the self; metamorphosis of the body; ethnographic research





## 1. Introduction

The cognitive path we will follow within the paper starts from the definition of embodiment to show how this concept can find concreteness in political co-communication. The paper concludes with a methodological proposal for the study of this phenomenon, namely through the application of netnography. In the first section, we will deal with the concept of digital embodiment; the latter, being polysemous should be de-emphasized in the meaning given in this paper. In particular, this concept is understood as the ability to embody emotions within an avatar through a process of redefining emotions, thus not a simple transposition, re-establishing new characteristics and new modes of interaction. This leads to, as a direct consequence, the construction of the self in a communal manner, that is, the co-construction of the self. Within the second section, we will go into the specifics of an example of embodiment in the context of political communication. In particular, we will discuss the use of such a feature by a political subject to initiate a process of change in political communication, with the aim of reaching a larger audience. To finish, in the last part of the article, we will specifically propose a methodological approach useful for studying the processes of embodiment. In particular, we will analyze how netnography can be a useful tool for studying the processes of self-construction in another world.

## 2. Digital Embodiment as the New Frontier of Co-Construction of the Self

The concept of embodiment has a polysemous character; most of the time, when one hears about this concept, the first thing that comes to mind is a clear reference to religion, particularly the concept of incarnation or reincarnation. However, this concept does not refer to the sacred/profane, depending on which side you read the phenomenon,

but rather to the incarnation of one person with another. It has recent origins and is reconfigured within the perspective of the formation and manifestation of the self, or rather, the co-construction of the self.

In addition to this, the concept of embodiment can be reconfigured within different aspects. In fact, the earliest discourses on the concept, though obviously distant from today's interpretation and the one we will give within this article, date back to 1994, thanks to Deborah Lupton's studies in the article "The Embodied Computer/User" [1]. The author analyzed this concept from the user experience, basing her reflection on the time of use of a device and the degree of freedom this device is able to grant.

In recent years, the concept of embodiment has undergone drastic changes from the past. Previously, embodiment with a computer meant recognizing oneself in the codes or developments of simple websites or video games. Today, however, embodiment occurs in a variety of ways, including via avatars or even holograms, which offer large degrees of freedom within virtual contexts. Robotics and prosthetic devices have also opened up new possibilities for embodiment with machines. Indeed, prosthetics, which are increasingly precise in their movements and have constant software updates, some of which have haptic feedback that simulates physical sensations ever more accurately, have enabled deeper inter-actions between people and machines. From these concepts, more and more scholars have approached this theoretical framework, including cognitivists and, only in the last 30 years, the social sciences. From these concepts, more and more scholars have approached this theoretical framework; for example, there is work by cognitivists and, in the last 15 years, by social scientists. For decades, it was thought that classical cognitivism was the only way to explain cognitive processes. Among the best-known works are those of Shapiro [2,3], Wilson [4], Foglia and Wilson [5], Caiani [6], and Bermúdez [7]. However, these concepts are outdated and considered a psychological dysanalysis. So much so that, in recent years, sociology has been pushing to reflect on a renewed corporeality, especially in relation to robotics and artificial intelligences in mechanical bodies, developing insights and critical considerations from the processes of digital interaction to the processes of identification and recognition in avatars. The ever-increasing presence of artificial intelligences capable of synthesizing information, statistics, and analysis on engagement is a very interesting aspect that can hold together several aspects, including the emotional compartment. In fact, in the specifics of this work, the concept understood here of embodiment refers to the identification, not only of the "physical body," but especially of the self, within an avatar in the digital world. To be as clear as possible, it is the cultural, emotional, and practical co-construction of the knowledge and specificities of others by a physical person being re-structured, re-read, and re-interpreted through an avatar in the digitized world. Many scholars, until now, have limited themselves to reading this phenomenon as a simple rethinking of corporeality, even a transposition of the phenomena of the physical world into the digital world. However, this is only its proximate consequence. In fact, as one might mistakenly think, it is not so much the new corporeality that is fundamental but the re-structuring of the self, that is, its manifestation within a digital world. The latter element is increasingly based on the co-construction of the self and experiences (Simmel's Erleben) in encounters with other avatars. So, it is possible to see a physical body connected intrinsically to a virtual body, which synthesizes its actions, expressions, characteristics, emotions, and culture, re-reading them within another world, making the main user adapt. For example, Belk talks about the extension of the self in the digital world; he spoke about it in both 1988 and 2013, and, like Lupton, at that time, he relied on users' experience of using computers.

What is of extreme interest in this paper is the update to Belk's article that occurred in 2013. In fact, the author added the concept of the avatar as an extension element of corporeality, a slightly different concept from embodiment but one that has several affinities. The first link is found in the reflection on the possession of objects in relation to the construction of the self. In this regard, the author specifies that what transforms the self, or the various selves (as he tends to emphasize several times), are a person's

various possessions. One tries to own as many things as possible, perhaps even expensive things, in order to modify oneself in the presence of other people. We could compare it to digital boasting. Moreover, compared to the 1988 article, in the 2013 article, there are five elements that characterize the extended self within digital processes, which are: (i) Dematerialization; (ii) Re-embodiment; (iii) Sharing; (iv) Co-construction of the self, and (v) Distributed memory [8].

When Belk speaks of dematerialization, he makes a clear reference to the dematerialization of what one owns. An example would be photographs; at one time, these were printed, but today, they can be conveniently stored in the cloud. It is the same situation with letters and postcards, as well as video games, which are progressively being transformed from a physical disk into digital files.

The second element Belk emphasizes is the process of digital reincarnation. This process, while not fully agreeing with the term used, represents the ability to reincarnate into an avatar. Before moving on to the detailed analysis of this element, it is necessary to point out that this concept could be identified as embodiment since the act of identifying oneself in an avatar is not a permanent situation, or at least it is not as permanent as it has been in the past.

The third element that Belk highlights is sharing. Within his text, we notice how there is an important focus on sharing on the Internet, particularly the immense pool of content that can be found. This element is important because it gives Belk an opportunity to analyze the concept of the individual self and the aggregate self. In particular, he tries to analyze how these two elements can improve so as to pose differences between yesterday and today.

The fourth element that Belk emphasizes concerns the co-construction of the self. As the author himself points out, the Internet is a place of sharing and meeting wherever we are invited to do so. Even in less likely places, such as on Spotify, a music listening app, we can share our playlists and receive likes. However, the world of video games has also changed dramatically. There are almost no offline games or games that involve single players; even soccer games, in which, until a decade ago, the online section was considered an optional extra, have now become a necessary part of the game. It is possible to acquire new languages that will become part of our personality, as happens in role-playing games such as the aforementioned World of Warcraft or League of Legends. Thus, a self that is built together with others according to Belk's idea and strengthened through what is related to that particular world. As a result, we find a shared self characterized by individuals with similar personalities and, above all, very similar tastes to each other.

Finally, Belk also includes the concept of distributed memory. This concept is directly related to what is highlighted in the co-construction of the self and retraces the element of autobiography, which is no longer unique but multiple and decentralized. For example, we will have one self in the physical world, one within video games, and, why not, one in dating apps. Everything, as Belk pointed out, is decentralized; that is, it no longer belongs only to our memory but is stored on physical or cloud media, so it can last "forever".

However, one could think of expanding Belk's general idea by updating it with elements belonging to the metaverse. In the network, in addition to the classic concept of co-construction of the self and shared memory, there is also the element of co-construction of emotions and feelings. On the net, there is no sharp distinction between what is right and what is not right. There is certainly some limitation and discouragement through prohibition about doing certain things, but in essence, there is no sharp separation. This causes more or less "factions" to arise at every opportunity.

Returning to the discussion earlier, several sites guarantee the possibility of getting to know a person in a short time, allowing fleeting encounters or forms of relationships other than those we tend to have. This leads to the emergence of new emotions, such as fleeting pleasure, or "menu" pleasure; this feature, surrounded by ad hoc techniques and language, leads to sharing with others. The latter element allows the standardization of some practices. For example, on Instagram, more and more minors are using the "Confession" or "Secret

Questions" feature in their stories; the average user will write, most often, things about intimacy and sexuality. This will lead to a repost of the comment by the user, with an attached response, bringing new knowledge and new forms of libido, even among the very young. Still other sites, as we shall see in the next section, allow for "political debates," translated into digitized election campaigns, mediated by avatars, embodying people, emotions, and ideas in a digital avatar, perhaps distant in physical characteristics (as we shall see in a few lines) but much more pregnant in manner. Above all, we find Second Life, which enabled Antonio Di Pietro to campaign (see next paragraph). However, one could also consider adding three sections to Belk's idea: emotional embodiment, social embodiment, and aesthetic embodiment. They relate perfectly to the metaverse, namely:

The "emotional embodiment" allows the person to fully identify with the avatar by experiencing feelings, perhaps new ones compared to those they have generally known. These feelings are enacted through in-game actions via the avatar. So, emotions start from a "physical" person but have a hundredfold scope in the avatar, which leads to a fleetingness of the moments experienced.

The "social embodiment," which gathers content related to the structuring of a character who must be cool within the virtual game, receive interactions, and, most importantly, generate a level of embodiment such that he or she has a voice within the team to which he or she belongs.

The "aesthetic embodiment" represents the players' willingness to build a character that lends itself, aesthetically, to be what it is not in reality. In this way, as was the case in the past with Second Life and as is the case today in PolkaCity or the Roblox metaverse, the degree of involvement is high.

Second Life is an online virtual environment launched on 23 June 2003. This environment constitutes an information technology platform in the field of new media and integrates a wide range of communication tools, both synchronous and asynchronous. In addition, it has found application in various creative fields, including entertainment, art, education, music, film, role-playing games, etc.

PolkaCity is a virtual game on blockchain that gives users ownership of virtual places and, with it, a steady source of income. Its main feature is the building of an avatar and the ability to open stores or start businesses, just as was accomplished in Second Life through microtransactions, but, in this case, it does so through cryptocurrencies.

One example is the creation of an avatar. Several aspects concur within this process, from aesthetics to emotionality to appearance. In addition, the concept of role-taking explored earlier by Mead [9] and Goffman [10] returns powerfully.

Certainly, these two levels alone are not sufficient; however, they might be useful for a preliminary description of the phenomenon. The idea is to initiate studies on cyberbullying, with the aim of understanding to what extent the cyberbully can be fully embodied in the character he/she plays online [11]. In addition, it might be interesting to investigate the relationship between cyberbullying and sensitivity to sensory processing, a personal trait that makes life events more salient and/or stressful. In fact, "individuals with high sensitivity show abnormal reactivity and emotional sensitivity to external and internal stimuli that, in turn, may interfere with daily life" [12] (p. 1).

This process is also closely related to another concept, which concerns emotions and is directly related to the physical user. The process, which we will call for simplicity, emotional embodiment, is a condition that occurs when a physical person interacts in a digital world and experiences emotions. From 2000 onward, there were numerous platforms that allowed the creation of an avatar for the purpose of connecting different people, even before the social boom, for example, Habbo Hotel and Second Life [13]. The first is presented (as still active as a community) as a virtual hotel, where different avatars, called Habbo, can interact with other users in private rooms or in common areas of access to all users of the site. The second platform is a virtual world, launched in 2003, which integrates synchronous and asynchronous communication tools and can be used as entertainment, art, education, music, and business. Also, in this world, we find avatars who are free to roam the world

and make new acquaintances. The latter, representing the precursor to the metaverse, is no longer the focus of users' attention. In fact, as much as both platforms use the same basic concept of creating an avatar and communicating, unlike the former, Second Life turns out to be much closer to the metaverse. In fact, using the same logic, many users have preferred to move to different platforms, such as Meta's Horizon World or PolkaCity. Thanks to the viewer, in fact, the activity performed through a computer on Second Life can be fully integrated into these other worlds. Moreover, many users using special prosthetics will even be able to have sensoriality of what takes place in that world. In addition, with the ability to access an open world, it will be possible to perform everything that tends to be accomplished on other sites, such as dating. Today, we find different platforms, starting with the metaverse Roblox or simpler sites like sMeet, that aim to bring avatars together and form bonds.

Understanding this phenomenon requires going beyond classical patterns and reflecting on the embodiment of emotions in virtual or virtualized bodies. A person's ability to correctly synthesize and describe his or her feelings and reflect them through interaction with others is what intervenes in this process, with the concept of emotional intelligence at the center of the discourses. The latter aspect generates a strong impact both in emotional training, face-to-face, and in the virtual compartment with embodiment. Interestingly, several analyses, such as the one conducted by Flavián, Ibánez-Sánchez, and Orús [14], highlighted the positive impact of technological embodiment through virtual reality. These devices were able to evoke more positive emotional reactions and higher psychological and behavioral engagement than computers and cell phones. Emotions and psychological engagement play an important role in mediating the impact of embodied virtual reality devices on behavioral engagement. This demonstrates the importance of technological embodiment in immersive experiences, such as in hotels that incorporate virtual reality into their communication strategies.

Therefore, specifically, embodiment is a construct through which the Cartesian mind-body pair is overcome to determine how humans see themselves as persons and how the mind interfaces with the body. Embodiment predicts that the individual self is the derivative of the integration of the representation of the social self and the physical self [15]. Indeed, people are social beings whose lives depend on the ability to predict and understand the emotions and intentions of others. They also depend on the ability to interpret the intentions of others by recognizing the specificities of others,. Many studies on the mirror neuron system (MNS) have demonstrated the activation of the same neural correlates underlying the observation of action in the third person and the execution of the same action in the first person; this neural resonance can be extended to social behavior [16–22]. Specifically, the idea is to analyze, through social experimental research, the scope of actions, for example, grouped into the three macrocategories added above Belk's idea. Specifically, one could start with the idea that the mirror neuron, a neuron deputed to imitation (and not empathy as emphasized by several neuroscientists), allows one to learn new patterns of actions and replicate them, not only in the "physical" world but also in the "online" world. So, thanks to the concept of "recognition of the specificities of the other" it becomes possible to replicate actions of the "physical" world within the "online" world mediated by the avatar. Therefore, as the "online" world is frenetic, these take on a very high scope, so much so that they become "new emotions" that are totally detached from the known ones, allowing the avatars to "tune in".

In conclusion, the idea is that Belk's idea needs further expansion, as there has been a radical transformation of information systems and digitization of lives since 2013. So much so that it has led to confusion between the "offline" world and the "online" world; for example, the new augmented reality and virtual reality systems installed on glasses or cars allow us to be perpetually between two worlds. For this reason, we should also talk about emotional embodiment, social embodiment, and aesthetic embodiment in order to give a broad and accurate description of the embodied world we are experiencing.

### 3. The Metamorphosis of the Body: The Central Role of Personalization and Engagement in the Political Sphere

What we have seen so far is useful as a starting point for analyzing another concept, namely that of the processes of embodiment within the contemporary political sphere. Indeed, just as there has been a radical change in the processes of digitization of lives, there has been a process of transformation of bodily presence and languages in the realm of politics. Indeed, an interesting line of inquiry emerges on the interconnection between political communication and the conceptual and visual metamorphoses of the networked body politic [23,24]. This section intends precisely to investigate the intricate transfiguration of the political corpus, focusing on the inherent intertwining of the concepts of personalization and participation. After all, political and communicative evolution increasingly embraces personalization strategies aimed at shaping the image of leaders and parties in order to catalyze consensus and influence public opinion [25–29].

The combination of personalization and participation represents a significant phenomenon in contemporary political communication, emphasizing the convergence of two crucial dynamics: the adaptation of political messages to the specific characteristics and preferences of individuals (personalization) and the active encouragement of individuals to participate in the political process (participation). Initially, the discussion of personalization emphasizes the increasing ability of political campaigns to tailor messages to voters' specific preferences, beliefs, and interests [30]. Such personalization is often facilitated by the vast amount of data collected through digital platforms, enabling the creation of highly targeted content [31]. However, it is crucial to note that in the context of this personalization, questions emerge regarding the manipulation of public opinion through selective filtering of information [32]. Next, the discourse shifts to participation, emphasizing the active invitation to individuals to engage in the political process. Digital platforms provide interactive channels that incentivize participation through opinion sharing, participation in online polls, dissemination of political content, and mobilization for specific causes [33]. It is, however, crucial to consider the risk of polarization and the effect of "echo chambers" in which people participate primarily with like-minded individuals. The importance of this combination stems from its ability to create a dynamic interaction between citizens and political actors, allowing for more direct and personalized engagement. This approach can increase the effectiveness of political communication, bringing individuals closer to decision-making processes and stimulating a sense of civic ownership and responsibility. Equally, it should be noted that the misuse or distorted use of these practices can also undermine the quality of public debate and fuel polarization. Therefore, critical understanding and analysis of the ethical implications of this combination are essential to fully assess its impact on the democratic sphere [34,35].

Personalization, then, manifests itself as a key element of contemporary political communication, in which leaders and political parties adopt communication strategies that are strongly oriented toward building a personal brand [36]. It should be emphasized, however, that the rise of the trend toward personalization constitutes an inherent feature of contemporary democracies, and following this trajectory is a decline in the importance attached to political programs and parties [37]. Hand in hand with this occurrence, moreover, there is a concomitant devaluation of the traditional role of political parties, found in the evident decrease in the level of trust that the electorate has in them [38]. Indeed, political identification, which was once closely associated with party affiliation, appears to be undergoing a substantial transformation. A relevant phenomenon in this regard is emerging more and more prominently, namely that the orientation toward the political party is increasingly replaced by the orientation toward the charismatic leader, who assumes a prominent role in the individual's political identification [39].

This transformation involves the strategic use of language, images, and narratives that highlight the individual's distinctive traits, and in this state of affairs, the evolution of the political body is to be understood not only as a metaphor but also as a tangible representation of the leader's personality and characteristics. Gestures, facial expressions,

and even clothing can be considered communicative tools that contribute to building the image of the leader as an authentic and accessible figure [40]. A true presidential trend that responds to the absolute need to capture the fluctuating electorate, poorly represented by parties but attracted by a leader seen as intimately closer [41].

Current political leaders are those who demonstrate a superior ability to respond effectively to the growing inclination toward window dressing, a phenomenon that emphasizes the visibility of leaders' personal aspects in the political arena [42]. The shift toward the privatization of the political sphere represents a significant shift from the preeminent importance of traditional media to the increasing emphasis placed on social media. In this environment, it is observed that the attention of the constantly fluctuating public is now less focused on the administrative abilities of political leaders and more on their personal qualities, particularly the characteristics that make them closer to the everyday experience of citizens [43]. This transition is accompanied by a gradual decline in the formal political interest shown by citizens, as the focus is shifted from the evaluation of the political organization and its electoral program to a greater consideration of the politician's person. It is observed that such weighting on the figure of the politician contributes confers legitimacy not only to his or her personal characteristics but also to the values and political actions promoted by the public actor [44]. Moreover, it seems evident that it is the leaders themselves who push toward the personalization of politics by focusing on the personal qualities of affability and likability, convinced that consensus on an emotional level can overcome the resistance of a public unattracted to formal politics [45].

In parallel, the dynamics inherent in conventional media are undergoing significant changes in the current media landscape, defined as a hybrid ecosystem, as outlined by Chadwick [46]. Indeed, we are spectators to the integration of the logic inherent in new media that brings the introduction of new communication channels, newly designed formats, and innovative languages, all of which are taking their place alongside traditional media. This change implies a complex coexistence between the established conventions of traditional media and the new modes introduced by digital media. This convergence involves the expansion of the available media repertoire, offering audiences a wide range of options and communicative approaches. Undoubtedly, interacting with users of online platforms, identified as "networked publics," can carry the potential risk of suffering a deterioration or enhancement of one's reputation, as well as seeing one's perspective or proposition strengthened or defeated [47]. Digital media platforms have significantly expanded the scope within which representation of the political sphere occurs and interaction with a variety of audiences. Concurrently, there is a wide diversification of content sources within a continuum in which narrative and informational aspects converge.

As things stand, digital media serve as an extended stage where political dynamics unfold and where different segments of the public interact in the established continuity that characterizes the interaction between the physical and digital dimensions, in which citizens operate to express their opinions and manifest their feelings in a multidimensional reality of life that unfolds both online and offline [48]. This plurality of voices and viewpoints offers a diverse range of perspectives on politics, enabling a more comprehensive and articulate understanding of political events and issues. At the same time, it should be noted that the digital media ecosystem is characterized by a coexistence of narrative and information, where the distinction between the two can be blurred. This implies that while there is an increasing emphasis on narrative and engaging presentation of political content, it still becomes necessary to provide accurate and reliable information in order to ensure a comprehensive understanding of political events and issues of public importance.

So, digital media have expanded the arena of politics and public engagement, but at the same time, present the challenge of balancing engaging narrative with accurate information, creating a complex and dynamic environment for contemporary political communication. Indeed, a considerable amount of time has seen a shift from a form of democracy based on political parties, which acted as intermediaries on behalf of citizens, to a form of democracy centered on the public [49]. This transformation is based on a direct link, both political

and communicative, between political leadership and the citizenry. In the context of this evolution, traditional logics of representation have gradually transformed into logics of representation, characterized by an increased emphasis on the speed of decision-making facilitated by the mass media [50]. This shift reflects a significant redefinition of the way democracy is practiced and includes a shift from the mediation of political parties to direct interaction between leaders and citizens, with significant consequences for political communicative dynamics.

The strategies adopted in order to create an emotional connection between politicians and their supporters are based on the deconstruction of tangible elements that constitute the physical presence of politicians, transforming them into more abstract and conceptual entities. The approaches used involve a process of "dematerializing" the image of the politicians themselves. This process involves the transformation of politicians from concrete physical figures to more ethereal and abstract entities. In other words, it involves emphasizing the non-material and symbolic aspects of politicians rather than their physical presence. This type of scheme aims to overcome physical barriers and create a feeling of closeness and identification based on shared ideals, values, and symbols.

In this way, the focus shifts from the materiality of the politician's body to its symbolic representation and the meaning it carries. Given the current context characterized by high complexity and information saturation, this practice contributes to the transition to a politics that tends to pop [51–57]. It represents, in essence, a remarkable effort to simplify the complexity of the political environment for the citizen who seeks a clear guide to navigate political dynamics. On the other hand, within contemporary politics, there is an increasing turn toward symbolic representations of political personalities, behaviors, and physical appearance, as well as the creation of rules and regulations [58]. This is inscribed within a social context that can be described as increasingly liquid [59], characterized by considerable instability and fluctuating electoral processes [60].

This is imbued with a constant climate of permanent campaigning, a concept widely discussed by Blumenthal [61]. This term represents the significant transformation of political dynamics, in which politics is no longer confined to well-defined periods of campaigning but rather constantly permeates public life. For this reason, the involvement of highly skilled professionals in the field of communication becomes imperative, as numerous elements and strategies that in the past were considered extraneous to the political field now take on a critically important role.

The body assumes a crucial function in the process by which individuals define their social identity, and this process seems to be increasingly influenced by the accentuation of outward appearance as a significant component. In other words, physicality becomes a determining element in the construction of individual identity within the social sphere, in which attention to physical form and appearance gains increasing importance. The abstraction from the physical dimension of the individual in the virtual environment opens the door to a relentless process of identity elaboration, which is significantly accentuated through the use of social media. In other words, the transformation of the body into a digital form enables constant refinement and modification of individual identity, a phenomenon that is amplified by the presence of online sharing channels. This means that people are able to shape their identity continuously and dynamically through online interactions, reinforcing the crucial role of the virtual dimension in the construction and expression of personal identity.

Following the discussion conducted, it is of paramount importance to highlight a case that has greatly influenced the reformulation of interactions between the online and offline spheres, as well as the reinterpretation of the traditional concept of the body of the political leader. Antonio Di Pietro, who previously served as the leader of the party "Italia dei Valori", deserves special recognition for his anticipation in the use of communication tools in the Italian political landscape and beyond. The former magistrate can be considered a genuine forerunner of "social" politics even before the Metaverse era. In fact, on 12 July 2007, he held the first Italian rally on Second Life, demonstrating a pioneering vision in

the unashamed use of certain tools. His initiative represented a significant antecedent to contemporary political practices, anticipating emerging trends in digital politics and social media presence. Di Pietro demonstrated advanced insight into the growing importance of social platforms as a tool for political engagement and mobilization. More importantly, he is the first shining example of a process of digital embodiment in campaigning. Suffice it to say, in addition to the process of ubiquity that has become apparent, this process is a clear transformation, and, if you will, step forward, from election campaigns carried out on socials such as Instagram, TikTok, and Facebook. The association between Antonio Di Pietro's name and the Second Life experience may seem distant in the collective memory today, but reflecting on this anticipation, it is possible to conceive of a substitute imaginary in which Second Life is replaced by the more current concept of the Metaverse and in which contemporary politicians follow Di Pietro's example, adapting to the evolution of the means of communication and political participation in the digital age. So, political debates between avatars whose embodied emotional compartments would turn out to be more charged than the classic debates. In this sense, he can be considered an enlightened precursor of digital embodiment in politics, whose contribution has played a significant role in shaping today's political landscape. This circumstance highlights his ability to virtualize physical presence, that is, to transform bodily identity into a digital presence, thus helping to redefine the paradigm through which politicians can interact with and engage citizens.

As early as July 2007, the digital and web worlds laid the groundwork for what would become a contamination of political debate, hinting in advance at a strong personalization and accentuation of aesthetic form. For this reason, it is not surprising that today, politicians have long been protagonists in a process of adaptation to digital technologies that has entailed a transformation of media and audiences [62,63], even making the times when they were reluctant to land on social networks appear outdated [64]. Nowadays, however, a case of particular interest emerges that could be seen as the culmination of the further dematerialization of the figure of the political leader.

This contingency is an illuminating example of the technological innovations and new engagement strategies adopted by politicians to interact with the public in an increasingly digital and interconnected context. What might appear to be the last frontier of evolution within political communication, namely the recent entry on the Chinese TikTok platform [65], already seems to have been surpassed. In France, one could observe the use of election rallies held in the form of holograms, an event that constitutes the apex of a process of increasingly marked dematerialization of the figure of the political leader. This phenomenon represents an emblematic example of the adoption of advanced technologies in the political arena, in which the political leader is virtualized into a digital representation projected to the public. Such innovation embodies the continuing evolution of political communication strategies, highlighting the importance of understanding how public perceptions and political participation are being altered in a context of increasing hybridization between the virtual world and physical reality.

This development suggests that innovation in political communication is moving rapidly toward the use of new media and advanced technologies, and the use of holograms in election rallies represents a further extension of the bodily dissolution of politicians as it allows them to interact with the public in a virtual and immaterial way. The event reflects a substantial advance in the way politicians seek to connect with the electorate and influence political dynamics in the digital age.

Indeed, on transalpine soil, radical leftist candidate Mélenchon, in the last French presidential elections, addressed the public in 12 different cities simultaneously, focusing on issues of popular sovereignty and globalization through the use of digital technologies to project his own hologram on numerous stages across the country. The leader of La France Insoumise was physically in Lille, a traditional leftist stronghold in the nation's north, while 11 other projections of his body trod the stages at as many rallies around France. This was not the first time for the politician, having already employed this hi-tech gimmick in the past 2017 presidential elections, appearing on two stages (Lyon and Paris).

In the first round, outgoing President Emmanuel Macron (LREM) achieved 27.85% of the vote, compared with 23.15% obtained by Marine Le Pen (RN), only in third place Jean-Luc Mélenchon (LFI) with 21.95%. Beyond the epilogue that relegated the leader of the radical left to third place, the result in the first round of the presidential election represents the best peak of his career, thanks also to the technological tools used [66].

In addition, another figure to be strongly considered is that of Mélenchon as the candidate most voted for by the 18–24 (31%) and 25–34 (34%) age groups. The size of the impact is still difficult to quantify, but certainly, one can appreciate the attempt to shorten the distance between politicians and citizens, especially given the high rate of abstentionism. On the other hand, the continued growth in the number of people abstaining from voting understandably constitutes an element of concern with respect to the degree of legitimacy of the representative system. Indeed, election results in Italy as well as in France show an unstoppable growth of the "party of non-voting," considered the most common option in this latest round of elections.

So, while the interaction between politics, digital communication, and new technologies raises legitimate questions and concerns about the quality and real effectiveness of civic engagement, we cannot overlook the innovative potential represented by the techniques that have enabled a dematerialization of the figure of the political leader. Before concluding, it is interesting to argue that these elements, if properly exploited, could not only improve citizens' enjoyment of political content but also reinvigorate interest and active participation in political life. Therefore, it is necessary to reflect on a methodological hypothesis for continuing to explore and deepen political practice in the digital world. In this regard, these opportunities could be used to promote greater civic mobilization and create a more robust sense of ownership, thus helping to shape the future of democracy in the digital age. So, in the next section, we thought we would propose a methodological idea for the careful analysis of these issues.

## 4. Ethnographic Research for the Study of Digital Embodiment

In this section, we want to emphasize the centrality of the ethnographic approach in the study of the interactive dynamics that take place in the digital dimension.

Ethnography was born as a method of knowing otherness in all its forms and manifestations: the process of translating cultural diversity and the rules underlying 'other' cultures forms the basis for ethnographic work. It is precisely cultures, through their representations, narratives, beliefs, and rituals, that are the specific object of ethnography, and the experience of confronting differences brings out the relativity of the contexts in which social practices take shape.

This is not simply a description of external reality, nor is it a matter of reporting facts in their veracity, but of paying attention to the social process of reality construction whose content is generated not only by the way that reality is produced but also by the relationship between ethnographer and the social context with which he or she enters into a relationship. It is a process that takes shape from what Colombo [67] calls the crisis of legitimacy, that is, from the recognition that all narratives and descriptions are not simply a snapshot of reality but contribute in some way to the creation of that reality: all of which certainly makes the ethnographer's authority problematic with respect to the version he or she gives of the reality studied. In this sense, the researcher's interpretation is not objectively valid, much less universally valid, but becomes one of the possible interpretations of that situation; his or her work refers to an interpretive process that is open and dialogic and cannot be evaluated solely for its completeness and variety of details or for its adherence or otherwise to reality. Interpretation remains a process that is the result of continuous negotiation with the field and the object of research [68].

The most common meaning with which we refer to ethnography is that of a predominantly empirical mode of knowledge based on observation, "translation", and writing, a definition that gives ethnographic practice a predominantly technical meaning linked exclusively to the scanning of its application phases and in some ways generic and therefore

adaptable to many situations. Ethnography is a widespread practice in many social sciences today, from sociology to anthropology, from social psychology to the study of organizations and pedagogy.

The ethnographic approach takes on a complex and multifaceted role in the framework of the social sciences of the twenty-first century; it is configured as that method that studies from the inside the ways in which social actors construct the world of everyday life. Doing ethnography is a style of research, and a particular way of observing, it is a particular way of participating, which always begins with a particular way of "looking around" (Quarta, 2020 [69]).

As Dal Lago makes clear, "looking around" is the metaphor frequently used to describe the ethnographic style [70]; it is a metaphor that only partially expresses the complexity, richness, and variety of actions contained in the ethnographer's work.

Thus, there is an ethnographic style of fieldwork that we can identify in a number of elements that give ethnographic research itself a scientific status: in its way of observing and describing social practices, in its not claiming to be exhaustive or to objectively capture reality; in its ability to come to identify invariant or generalizable types of practices, while starting from partial points of view; in its not being a simple narrative, never completely naive [71,72].

In order to convey his own experiences and make others' experiences his own, in order to make himself understood and to understand the other, the observer cannot remain imprisoned in his own system of scientific disclosures and typifications; if he did so, as Garfinkel [73] showed in his now classic experiments, he would risk incommunicability by anomie, meaninglessness. He must, therefore, abandon the system of scientific relevance and typifications to make room for that of the world of everyday life. That is, he must leave the world of sociology, the world of contemporaries, and enter the world of the environment, where there is spatio-temporal immediacy and the possibility of communication.

Rather than developing a sociological theory in which it is possible to explain social reality at the macro level to understand institutions from the social actor and his interactions, ethnographers study the method used by the social actor to "construct" the world through actions based on his own interpretations.

Ethnography could not remain indifferent to the metamorphosis of social relations, and several scholars [74–77] have begun to pay attention to the significance of digitalized spaces as environments for living, interacting, and co-constructing social reality.

The exponential rate at which network use has spread, as we have seen in the previous paragraphs, has inevitably affected the way we construct our identity and the social reality around us. The online community has the connotations of fluidity because person and identity, structure, and time are different, and the channel through which they are manifested is also different.

With the spread of the Internet, the spaces of the social sphere have found a proliferation in the world of the three double 'w's, becoming new habitats in which subjects create and reproduce relationships, identities, and social places.

The Internet is now not only a technology but has become an engine of social change, which has altered many of the habits of individuals, from social relationships and work, to the way we learn news and engage in work activity, and perhaps most importantly, it is also influencing the way we construct hopes and dreams [78]. The Web is not only to be considered as the main source of our information so much so that it contributes to the definition of today's society as the information society; in fact, this medium has become much more: it is a social space, an environment, made and enabled by communication (the cornerstone of community and society) [79].

Digitalized spaces and the interactions within them are no longer at the margins of social life; they have radically transformed the way people all over the world go about their daily activities, becoming central to the experience of real life: e-mail, WhatsApp, Facebook, Twitter, Instagram are just a few of the new ways through which to represent oneself and relate to others [80,81]. Online interactions find more and more space in the experiences of

daily life: it is a dimension that has become so widespread and influential in everyday life that it is no longer considered virtual but real for all intents and purposes [82].

"Virtual reality" is no longer a reality separate from other aspects of human action and experience but rather a part of it. The work of ethnographers has increasingly gone in this direction by including the Internet and the networked world in its epistemological and methodological scope in order to arrive at a more consonant understanding of all the determinants of current social life.

The area of study of networked ethnography, over time, has defined boundaries by including the social effects of internet use [83–85], the role of the Internet in everyday life as a crucial part of communication processes and interpersonal relationships [86–88], even going so far as to address the role of emotions on online communication [89–91].

As lived realities increasingly include online interactions, ethnographers studying contemporary social life should consider online spaces as another dimension in which their participants live; in addition to the natural habitat, they need to include the online habitat in which relationships find a new location and a new system of mutual influences [92].

The multiplication of terms referred to when talking about the application of ethnography in online communities (digital ethnography, virtual ethnography, ethnography on the Web, etc.) indicates that within the scholarly community, there arises a need to distinguish online ethnography from the classical approach that prefers the face-to-face relationship between researcher and studied context.

Observing social interactions through a PC screen or in the subject's presence by interacting directly in the physical places of interaction presupposes a profound change in the practice of ethnography.

Kozinets [93] wanted to denote this new field of application of ethnography with the neologism "netnography" referring to the study of communities on the Web. This new use of ethnography is simpler and less costly than traditional ethnography, as well as less intrusive than focus groups or interviews. It provides information about the symbolism, meanings, and consumption patterns of online consumer groups.

In its broader applications, netnography enables the study of communities and cultures that emerge from communication used in the digital environment.

In the large and ever-evolving system of social and individual data and communications that are generated on the Web, netnography lies somewhere between the vastness of big data analysis and the close readings of discourse analysis.

In recent years, many anthropologists, sociologists, and qualitative marketing researchers have written about the need to specially adapt existing ethnographic research techniques to the many cultures and communities that are emerging through online communications [94,95].

Netnography studies online communities that are obviously no less "real" than "physical" ones. Originating as a useful tool for marketing needs through the study of communities in which consumers have discussions about products and related brands, the method has also been applied in the social-psychological field, opening up new areas of analysis that include new ways of structuring relationships and self-representation across the many communities and cultures that populate the Web [96,97].

It is a methodological approach that proves particularly useful for studying the mechanisms that characterize the interactions of an avatar in a digitized environment: netnography offers the possibility of looking from within at the processes of cultural, social, and emotional co-construction that characterize the digitized environment. In this way, scholars can not only enter the dynamics of the construction of digital corporeality but are also able to analyze the process of re-structuring the self that comes to life from encounters with other avatars.

Enacting an observation in this environment means dealing with and managing a number of elements that are markedly different from conducting an in-presence observation in which relational space is managed by the proxemics of bodies.

The data resulting from a netnographic analysis can be rich or very delicate, protected or freely provided. They can be generated by a person or a group or co-produced with machines or software. They can be generated through interactions between a real person and a researcher or be stored in digital archives. They can result from a high level of interaction, such as a conversation, or they can arise from reading an individual's writing (such as a paper or a blog).

Netnography is a practice in which the researcher does not exclusively observe or analyze words but examines images, audio files, and drawings or may examine website creations or other digital products. Just as in research conducted in presence, the scholar, even in a virtual environment, can practice a level of participatory interaction through the application of interviews to be performed online, using appropriate platforms such as Skype or Google Meet.

Maintaining methodological rigor, punctuality, and focus, netnography offers a range of possibilities for understanding the production of social dynamics in which an avatar may be involved. It is a dynamic that requires the interpretation of human communications in realistic contexts, in situ, in the native conditions of interaction, when these human communications are shaped by new technologies.

It is a type of observation that we can liken to the non-participant observation that Spradley [98] talked about when he referred to the research experiences that could be had by watching television programs; although we are talking about different contexts, the similarity arises from the position of the researcher who stands at the edge of the field of inquiry and his presence cannot be revealed in any way by the people being observed.

There are advantages of the peculiarities of netnography that make it very interesting as a new frontier of application of the ethnographic approach in an environment where the self is reproduced digitally. First of all, access is much easier to put in place; having less intrusiveness than an in-person observation cancels out the problem of responsiveness, among other elements, and also makes it possible to carry out the observation in a time set exclusively by the researcher. It can also provide information in a less costly and more timely manner compared not only to participant observation but also to focus groups and personal interviews.

This is also compounded by the fact that it can use digital approaches such as social network analysis, data science and analysis, visualization methods, social media research presence, and videography.

Applications and publications using netnography are flourishing in fields as diverse as geography, sociology, media studies, travel and tourism, sexuality and gender research, nursing, addiction research, game studies, and education.

These are research studies that investigate the different areas within which transformations of those practices that contribute to the construction of social reality are most evident. I am referring to the research on the influence of social networks in the way social identities and social relations are represented, from the political to the economic sphere; to the investigation of the way young people experience their changes in status and define learning practices using the Internet and new media; to the investigation of the dynamics of approval and social recognition carried out through the analysis of likes on Facebook [99–101].

The path of reflection outlined in this section allows for a consideration that links digital embodiment and netnography.

On the one hand, digital embodiment finds in netnography one of the most important tools of investigation: the interactions of an avatar are characterized by actions and reactions within an environment in continuous (social) construction; in this context, the method has the ability to exploit the circumstantial interpretative stimuli that, within a dialogical process, bring out the deep connotations of digital embodiment.

On the other hand, in the study of digital embodiment, netnography finds a fundamental test bed for refining its potential and testing research tools that, applied in this new frontier of digital reality, contribute to the emergence of new reflections and allow the

effective analysis of virtual life contexts, making them appear as an inexhaustible source of knowledge on the processes of construction of social identities, their relations and the common sense that underlies them [102–104].

## 5. Conclusions

The concept of embodiment appears to be a new frontier of self-representation, which, since Mead, we have understood as an element capable of constructing and not transposing new interactive processes, but more importantly, thanks to embodiment, new selves are co-constructed through the process of erlebnis, that is, of lived experience. Therefore, it is necessary to find and experiment with knowledge methodologies useful for analyzing the defining elements and processes of self-constructions within the meta-verse. Netnographic research seems to offer a useful tool for studying such mechanisms as they take place in virtual reality. Therefore, as pointed out, there is a double challenge; on the one hand, the study of embodiment can find in netnography an effective method of analysis; on the other hand, it is a challenge for the entographic approach to be applied in a completely, or almost unexplored, everyday life environment.

**Author Contributions:** Conceptualization—V.A., D.B. and S.Q.; writing, V.A., D.B. and S.Q.; review and editing, V.A., D.B. and S.Q. All authors have read and agreed to the published version of the manuscript.

**Funding:** This research received no external funding.

**Institutional Review Board Statement:** Not applicable.

**Informed Consent Statement:** Not applicable.

**Data Availability Statement:** Not applicable.

**Conflicts of Interest:** The authors declare no conflict of interest.

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
