# Peer review of "Digital Embodiment as a Tool for Constructing the Self in Politics"

_societies, doi:10.3390/soc13120261_

Round 1

Reviewer 1 Report

Comments and Suggestions for Authors

Review Digital embodiment as a tool for constructing the self in politics

Abstract:

The abstract starts very clear. Only in the last sentences, where I wanted to read the main message, it becomes a bit messy. It would be great if you could make a more clear statement here and possibly reduce the times you use the words ‘methodology’ and ‘approach’.

Section digital embodiment…:

46-47: here you continue to the cognitivists and social science, however, the following sentences are only on the cognitivists, which made it for a moment confusing. Try to rephrase or introduce this part differently.

47: Is it really true that it is just last 15 years, since you mention Deborah Lupton earlier and her definition of 1994. This is contradictory.

57-61: Please try to rephrase this sentence.

You talk about Belks table, but this is not shared with the reader. Maybe delete the reference towards this table or add something to make it visual.

The discussion of the fourth element is out of proportion (especially compared to the others) and would be better to shorten. Take only the main elements relevant to your work and if the reader is interested they can search for the article of Belk.

147-148: Please directly mention the two sections here before getting into discussing them. And why is the emotional embodiment not part of these two (thus three)?

The explanation of Second life is quite extensive and I do not see the added value of most details described here.

223: In the end of this section you come with MNS, but compared to everything above you do not very much get into detail about this. There is a bunch of references here and it feels strange compared to the previous sections to not explain more about this.

Connected to this, and an overall comment to this section, is the missing aim or objective of your manuscript. It seems that you want to adjust the table of Belk?

Section the metamorphosis…:

229: Here you start with the political sphere, and maybe this is connected to the aim of your manuscript? You mention multiple aims in the first two paragraphs here.

229: Please try to connect this section more towards the previous section before starting this sentence. Now I am missing a connection and why there is ‘also’.

In the first paragraphs of this section you make a switch more to the political. Which was expected in your manuscript, but not there in the previous section. It feels a bit disconnected from each other and why do you not make some ‘political sphere’ statements in the previous section? Especially since in this section it is very specific regarding parties and leaders.

237: You mention the combination of personalization and participation. About the first concept you start very clear and I have the feeling the participation is more in the second part, but please make more clear when you make the switch from discussing the personalization and start discussing the participation. Also how is this combination important?

312: Due to the ‘in sum’ it seems you are ending the section.

371: I really like this example it is does justice to your manuscript. One thing that would make it even better would be a discussion connected to this about the appearance of these political leaders in the virtual world. Would this give them a ‘different’ body, a more digital body, and how would that influence their parties or the citizens.

Overall about the second section, you make very interesting arguments, but I would like to see more connections with the digital embodiment concepts and the virtual or avatars as was discussed in the previous section.

437: The last paragraph of this section seems already like an ending of the entire paper. I assume with this you want to introduce the next section otherwise I want to ask you to remove this section towards the end of the manuscript. It sounds like recommendations, but also like a bit of the aim of the next section in which you come with the methodological approach.

Section ethnographic research…:

447: Similar to the start of the previous section, for me ethnographic approach falls from the sky and you do not give an introduction from the previous sections into this section. What is the research to search for an approach and what is the reason you consider the ethnographic approach. I think the reasoning why this is the most suitable will follow. Also I think a visible aim somewhere in the beginning of the paper would again make more clear.

You provide a clear, but long, explanation of ethnographic research. Which of these aspects are the most important for your work and the use of this approach in view of digital embodiment and the political sphere?

I will recommend you to focus more on the second part, netnography. This sounds interesting and relevant in view of the two earlier sections. Try to make more connections with these earlier sections and how the approach is useful and valuable.

Similar to the abstract the ending of your manuscript is much more unclear compared to the first two sections. What is/are main messages you will provide? What is the value of the methodological approach you recommend and how to go from here? Try to find an ending for the entire story and not only for this last section. For now I am wondering why to read the first sections and not only the last part…

General comment:

Overall, I think the manuscript is quite lengthy. I mentioned some parts already in my comments, but I would like to suggest to only keep the most relevant topics in the paper. Sometimes I loose sight of the overall message.

Also, the sections are very long which could maybe as well be improved with more guidance for the reader, more connections between sections and paragraphs or some sub-headings.

Comments on the Quality of English Language

The quality of the English Language is fine. Here and there some lengthy sentences which could be improved.

Author Response

Thank you for these revisions, we are glad we could improve our work. Below I list what we have modified:

In Abstract section we have included: 

Therefore, we will seek to analyze how the personalization of the body within political communication has been profoundly affected by the virtualization of human experience. Next, then, we will introduce a new approach, useful for studying this fusion, that can emphasize the importance of analyzing this issue using ethnography, which is useful for fully understanding the complex dynamics surrounding the personalized digital body.

In Digital embodiment as the new frontier of co-construction of the self we have included: 

From these concepts more and more scholars have approached this theoretical framework, including cognitivists and, only in the last 30 years, the social sciences. From these concepts more and more scholars have approached this theoretical framework, for example, there is work by cognitivists and in the last 15 years by social scientists. For decades it was thought that classical cognitivism was the only way to explain cognitive processes. Among the best known works are those of Shapiro [2] and [3], Wilson [4], Foglia and Wilson [5], Zipoli Caiani [6] and Bermúdez [7]. However, these concepts are outdated and considered a psychological dysanalysis. So much so that, in recent years, sociology has been pushing to reflect on a renewed corporeality, especially in relation to robotics and artificial intelligences in mechanical bodies, developing insights and critical considerations from the processes of digital interaction to the processes of identification and recognition in avatars. The ever-increasing presence of artificial intelligences capable of synthesizing information, statistics and analysis on engagement is a very interesting aspect that can hold together several aspects, including the emotional compartment. In fact, in the specifics of this work, the concept understood here of em-bodiment refers to the identification, not only of the "physical body," but especially of the Self, within an avatar in the digital world. To be as clear as possible, it is the cultural, emotional and practical co-construction of the knowledge and specificities of others, by a physical person being re-structured, re-read and re-interpreted through an avatar, in the digitized world.  

The second element Belk emphasizes is the process of digital reincarnation. This process, while not fully agreeing with the term used, represents the ability to reincarnate into an avatar. Before moving on to the detailed analysis of this element, it is necessary to point out that this concept could be identified as embodiment, since the act of identifying oneself in an avatar is not a permanent situation, or at least it is not as permanent as in the past.

The third element that Belk highlights is sharing. Within his text we notice how there is an important focus on sharing on the Internet, particularly the immense pool of content that can be found. This element is important because it gives Belk an opportunity to analyze the concept of the individual self and the aggregate self. In particular, he tries to analyze how these two elements can improve so as to pose differences between yesterday and today.

The fourth element that Belk emphasizes concerns the co-construction of the self. As the author himself points out, the Internet is a place of sharing and meeting, wherever we are invited to do so. Even in less likely places, such as on Spotify, a music listening app, we can share our playlist and get likes. But the world of video games has also changed dramatically, there are almost no offline games or games that involve single player, even soccer games, which until a decade ago the online section was considered an optional extra, have now become the necessary part of the game. It is possible to acquire new languages that will become part of our personality, as happens in role-playing games such as the aforementioned World of Warcraft or League of Legend. Thus, a self that is built together with others according to Belk's idea and strengthened through what is related to that particular world. And, as a result, we find a shared self, characterized by similar personalities to each other and, above all, very similar tastes to each other.

Finally, Belk also includes the concept of distributed memory. This concept is directly related to what is highlighted in the co-construction of the self and retraces the element of autobiography, which is no longer unique but multiple and decentralized. For example, we will have one from the physical world, one within video games and, why not, one in dating apps. Everything, as Belk pointed out, is decen-tralized, that is, it no longer belongs only to our memory, but is stored on physical or cloud media, so it can last "forever."

However, one could think of expanding Belk's general idea by updating it with elements belonging to the metaverse. For in the network, in addition to the classic concept of co-construction of the self and shared memory, there is also the element of co-construction of emotions and feelings. On the net there is no sharp distinction between what is right and what is not right to do, there is certainly a limitation and discouragement, through prohibition, in doing certain things, but in essence there is no sharp separation. This causes more or less "factions" to arise at every opportunity.

Still other sites, as we shall see in the next section, allow for "political debates," translated into digitized election campaigns, mediated by avatars, embodying people, emotions, and ideas in a digital avatar, perhaps distant in physical characteristics (as we shall see in a few lines) but much more pregnant in manner. Above all we find Second Life, which enabled Antonio Di Pietro to campaign (see next paragraph). However, one could also consider adding three sections to Belk's idea, emotional embodiment, social embodiment and aesthetic embodiment. They relate perfectly to the metaverse, namely:

The "emotional embodiment", which allows the person to fully identify with the avatar by experiencing feelings, perhaps new ones compared to those generally known, that are enacted, through in-game actions, through the avatar. So, emotions that start from a "physical" person but have a hundredfold scope in the avatar, which leads to a fleetingness of the experienced moments.

Second Life is an online virtual environment launched on June 23, 2003. This environment constitutes an information technology platform in the field of new media and integrates a wide range of communication tools, both synchronous and asynchronous. In addition, it has found application in various creative fields, including entertainment, art, education, music, film, role-playing games, etc.

Therefore, specifically, embodiment is a construct through which the Cartesian mind-body pair is overcome to determine how humans see themselves as persons and how the mind interfaces with the body. Embodiment predicts that the individual self is the derivative of the integration of the representation of the social self and the physical self [18]. Indeed, people are social beings whose lives depend on the ability to predict and understand the emotions and intentions of others. But not only that, also to interpret, by recognizing the specificities of others, the intentions of others. Many studies on the mirror neuron system (MNS) have demonstrated the activation of the same neural correlates underlying the observation of an action in the third person and the execution of the same action in the first person; this neural resonance can be extended to social behavior [19]; [20]; [21]; [22]; [23], [24]; [25]. Specifically, the idea is to analyze, through social experimental research, the scope of actions, for example, grouped into the three macrocategories added above Belk's idea. Specifically, one could start with the idea that the mirror neuron, a neuron deputed to imitation (and not empathy as emphasized by several neuroscientists), allows one to learn new patterns of actions and replicate them, not only in the "physical" world but also in the "online" world. So, thanks to the concept of "recognition of the specificities of the other" it becomes possible to replicate actions of the "physical" world within the "online" world mediated by the avatar. Therefore, as the "online" world is frenetic, these take on a very high scope, so much so that they become "new emotions" that are totally detached from the known ones, allowing the avatars to "tune in".

In conclusion, the idea is that Belk's idea needs further expansion, as there has been a radical transformation of information systems and digitization of lives since 2013. So much so that it has led to confusing the "offline" world from the "online" world, the new augmented reality and virtual reality systems installed on glasses or cars allow us to be perpetually between two worlds. For this reason, we should also talk about emotional embodiment, social embodiment and aesthetic embodiment in order to give a broad and accurate description of the embodied world we are experiencing.   In The metamorphosis of the body: the central role of personalization and engagement in the political sphere section we have included:   What we have seen so far is useful as a starting point for analyzing another concept, namely that of the processes of embodiment within the contemporary political sphere. Indeed, just as there has been a radical change in the processes of digitization of lives, there has been a process of transformation of bodily presence and languages, in the realm of politics. Indeed, an interesting line of inquiry emerges on the interconnection between political communication and the conceptual and visual metamorphoses of the networked body politic [26]; [27].   The combination of personalization and participation represents a significant phenomenon in contemporary political communication, emphasizing the convergence of two crucial dynamics: the adaptation of political messages to the specific characteristics and preferences of individuals (personalization) and the active encouragement of individuals to participate in the political process (participation). Initially, the discussion of personalization emphasizes the increasing ability of political campaigns to tailor messages to voters' specific preferences, beliefs, and interests [33]. Such personalization is often facilitated by the vast amount of data collected through digital platforms, enabling the creation of highly targeted content [34]. However, it is crucial to note that in the context of this personalization, questions emerge regarding the manipulation of public opinion through selective filtering of information [35]. Next, the discourse shifts to participation, emphasizing the active invitation to individuals to engage in the political process. Digital platforms provide interactive channels that incentivize participation through opinion sharing, participation in online polls, dissemination of political content, and mobilization for specific causes [36]. It is, however, crucial to consider the risk of polarization and the effect of "echo chambers" in which people participate primarily with like-minded individuals. The importance of this combination stems from its ability to create a dynamic interaction between citizens and political actors, allowing for more direct and personalized engagement. This approach can increase the effectiveness of political communication, bringing individuals closer to decision-making processes and stimulating a sense of civic ownership and responsibility. Equally, it should be noted that the misuse or distorted use of these practices can also undermine the quality of public debate and fuel polarization. Therefore, critical understanding and analysis of the ethical implications of this combination are essential to fully assess its impact on the democratic sphere. [37; 38].   So, digital media have expanded the arena of politics and public engagement, but at the same time present the challenge of balancing engaging narrative with accurate information, creating a complex and dynamic environment for contemporary political communication.  

In fact, on July 12, 2007, he held the first Italian rally on Second Life, demonstrating a pioneering vision in the unashamed use of certain tools. His initiative represented a significant antecedent to contemporary political practices, anticipating emerging trends in digital politics and social media presence. Di Pietro demonstrated advanced insight into the growing importance of social platforms as a tool for political engagement and mobilization. More importantly, he is the first shining example of a process of digital embodiment in campaigning. Suffice it to say, in addition to the process of ubiquity that has become apparent, that this process is a clear transformation, and, if you will, step forward, from election campaigns carried out on socials such as Instagram, TikTok, and Facebook. The association between Antonio Di Pietro's name and the Second Life experience may seem distant in the collective memory today, but reflecting on this anticipation, it is possible to conceive of a substitute imaginary in which Second Life is replaced by the more current concept of the Metaverse and in which contemporary politicians follow Di Pietro's example, adapting to the evolution of the means of communication and political participation in the digital age. So, political debates between avatars whose embodied emotional compartment, would turn out to be more pregnant than the classic debates. In this sense, he can be considered an enlightened precursor of digital embodiment in politics, whose contribution has played a significant role in shaping today's political landscape. This circumstance highlights his ability to virtualize physical presence, that is, to transform bodily identity into a digital presence, thus helping to redefine the paradigm through which politicians can interact with and engage citizens.

So, while the interaction between politics, digital communication and new technologies raises legitimate questions and concerns about the quality and real effectiveness of civic engagement, we cannot overlook the innovative potential represented by the techniques that have enabled a dematerialization of the figure of the political leader. Before concluding, it is interesting to argue that these elements, if properly exploited, could not only improve citizens' enjoyment of political content, but also reinvigorate interest and active participation in political life. Therefore, it is necessary to reflect on a methodological hypothesis for continuing to explore and deepen political practice in the digital world. In this regard, these opportunities could be used to promote greater civic mobilization and create a more robust sense of ownership, thus helping to shape the future of democracy in the digital age. So, with the next section, we thought we would propose a methodological idea for the careful analysis of these issues.

In Ethnographic research for the study of digital embodiment we have included:

In recent years, many anthropologists, sociologists, and qualitative marketing researchers have written about the need to specially adapt existing ethnographic research techniques to the many cultures and communities that are emerging through online communications [92]; [93]; [94].

It is a methodological approach that proves particularly useful for studying the mechanisms that characterize the interactions of an avatar in a digitized environment: netnography offers the possibility of looking from within at the processes of cultural, social and emotional co-construction that characterize the digitized environment. In this way, scholars can not only enter the dynamics of the construction of digital corporeality but are also able to analyze the process of re-structuring the self that comes to life from encounters with other avatars.

Maintaining methodological rigor, punctuality and focus, netnography offers a range of possibilities for understanding the production of social dynamics in which an avatar may be involved. It is a dynamic that requires the interpretation of human communications in realistic contexts, in situ, in the native conditions of interaction, when these human communications are shaped by new technologies [95].

There are advantages of the peculiarities of netnography that make it very interesting as a new frontier of application of the ethnographic approach in an environment where the self is reproduced digitally.   These are researches that investigate the different areas within which transformations of those practices that contribute to the construction of social reality are most evident.  

On the one hand, digital embodiment finds in netnography one of the most important tools of investigation: the interactions of an avatar are characterised by actions and reactions within an environment in continuous (social) construction; in this context, the method has the ability to exploit the circumstantial interpretative stimuli that, within a dialogical process, bring out the deep connotations of digital embodiment.

On the other hand, netnography finds in digital embodiment a fundamental test-bed for refining its potential and testing research tools that, applied in this new frontier of digital reality, contribute to the emergence of new reflections and allow the effective analysis of virtual life contexts, making them appear as an inexhaustible source of knowledge on the processes of construction of social identities, their relations and the common sense that underlies them [102; 103; 104; 105; 106; 107].

Reviewer 2 Report

Comments and Suggestions for Authors

The paper with the title: "Digital Embodiment as a tool for constructing the self in politics" is based on a significant hypothesis (?) (page 5): "Therefore, specifically, embodiment is a construct through which the Cartesian mini-body pair is overcome to determine how humans see themselves as people and how the mind interfaces with the body (...) Indeed, people are social animals whose lives depend on the ability to predict and understand the emotions and intentions of others"...Hard staff!

Since, the article is not structured in Introduction, Main Part and Conclusions, for me, it seems extremely difficult to sketch the argumentation line of the article. It seems clear, that the idea is, that digital technologies are able to transform the cultural concept of body. In the second part, the focus lies on the politicial sphere ("body") and the strategic use of "language, images, and narratives in ...the leader's personality" (p.6). I cannot understand the jump from new forms of the use of media in democratic societies towards the concept of "materiality of the politicians body"(p.7). What does this means? How is the transformation described? Afterwards there are (interesting) descriptions of empirical events in political contexts in Italy, France. But I cannot understand, the new embodiment within these descriptions. May be there are new type of discourses, new type of political events and new types of drawing attentions?

Afterwards there is ethnographic research for what? to show what? Also here are statements, which I do not agree: "Ethnography is the most essential form of social research because it does not seek the causes of phenomena but analyses the procedures of data construction as the result of negotiation amomg members of a society " (p.11). What does that mean? and how ethnographic research may function observing movements in the internet? "netnography" comes up "as a type of observation that we can liken to non-participant observation" (p. 12). Are there methodological interventions now?

With this chapter the article finishes.

Does the article deals about the phenomenon of "embodiment" in political context or does the article speak about ethnographic methods in the internet?

However, the article is nicely written and I feel that there is a lot of knowledge behind. The article has not structure, no clear argumentation line and no clear scientific focus. All of them should be re-developed in the next phase.

Author Response

Dear reviewer, thank you for these comments, they allowed us to improve our work. Specifically, it has been revised almost entirely. Starting with the abstract and ending with the paragraph on methodology.

Round 2

Reviewer 1 Report

Comments and Suggestions for Authors

Thank you for the revised version of the manuscript. I can see you considered the comments very well and improved the manuscript.

Author Response

Dear reviewer, thank you for your comment, we are happy to have responded to your inquiries.

Reviewer 2 Report

Comments and Suggestions for Authors

Dear Authors,

after reading the article I have recognised, that you have contributed important information in order to improve the argumentation line of your article.

However, as I described in my first evaluation, I really ask you to re-structure the article. There is no introduction, which orientates the reading person towards your intention or research question or observation, which leads to the objective of the article. Further more there is no orientation about the structure of the article, how to read it, how to understand it, how to deepen the idea of body in your context etc. etc..

Chapter 1) starts already with the different issues, which you are taken in account for your hypothesis (?). I would say, this is the theoretical/conceptional part of your article, which should be also be defined as such.

In sum: I would re-structure the article in order to provide orientation with regard to a) objective, b) research process and c) results of the article.

With kind regards,

the reviewer

Comments on the Quality of English Language

I am not an English speaking person. From my perspective, the language seems fine.

Author Response

Dear reviewer, thank you for your comment, we have tried to further improve the paper by writing an introductory paragraph that goes to analyze the whole course, explaining our intentions; and a concluding part that goes to propose an opening to a new line of inquiry. We hope to have answered your request. I insert below what has been added:

Introduction

The cognitive path we will follow within the paper, starts from the definition of embodiment to show how this concept can find concreteness in political co-communication. The paper concludes with a methodological proposal for the study of this phenomenon, namely through the application of netnography. In the first section we will deal with the concept of digital embodiment, the latter, being polysemous should be de-emphasized in the meaning given in this paper. In particular, this concept is understood as the ability to embody emotions within an avatar, through a process of redefining emotions, thus not a simple transposition, re-establishing new characteristics and new modes of interaction, this has as a direct consequence, the construction of the self in a communal manner, that is, the co-construction of the self. Within the second section we will go into the specifics of an example of embodiment in the context of political communication, in particular we will discuss the use of such a feature by a political subject, used to initiate a process of change in political communication, with the aim of reaching a larger audience. To finish, in the last part of the article, we will specifically propose a methodological approach useful for studying the processes of embodiment, In particular, we will analyze how netnography can be a useful tool for studying the processes of self-construction in another world.

Conclusions

The concept of embodiment appears to be a new frontier of self-representation, which since Mead we have understood as that element capable of constructing and not transposing, new interactive processes, but more importantly, thanks to embodiment new selves are co-constructed through the process of erlebnis, that is, of lived experience. Therefore, it is necessary to find and experiment with knowledge methodologies useful for analyzing the defining elements and processes of self-constructions within the meta-verse. Netnographic research seems to offer a useful tool for studying such mecha-nisms as they take place in virtual reality. Therefore, as pointed out, there is a double challenge; on the one hand, embodiment can find in netnography an effective method of analysis; on the other hand, it is a challenge for the entographic approach to be applied in a completely, or almost unexplored, everyday life environment.